# A Comprehensive Characterization of Stemness in Cell Lines and Primary Cells of Pancreatic Ductal Adenocarcinoma

**DOI:** 10.3390/ijms231810663

**Published:** 2022-09-14

**Authors:** Benedetta Ferrara, Erica Dugnani, Valeria Sordi, Valentina Pasquale, Silvia Pellegrini, Michele Reni, Gianpaolo Balzano, Lorenzo Piemonti

**Affiliations:** 1Diabetes Research Institute, IRCCS San Raffaele Scientific Institute, 20132 Milan, Italy; 2Department of Biotechnology and Biosciences, University of Milano-Bicocca, 20126 Milan, Italy; 3SYSBIO (Centre of Systems Biology), ISBE (Infrastructure Systems Biology Europe), 20126 Milan, Italy; 4Department of Medical Oncology, IRCCS San Raffaele Scientific Institute, 20132 Milan, Italy; 5Pancreatic Surgery, Pancreas Translational & Clinical Research Center, IRCCS San Raffaele Scientific Institute, 20132 Milan, Italy

**Keywords:** PDAC, stemness, pancreatic cancer stem cells

## Abstract

The aim of this study is to provide a comprehensive characterization of stemness in pancreatic ductal adenocarcinoma (PDAC) cell lines. Seventeen cell lines were evaluated for the expression of cancer stem cell (CSC) markers. The two putative pancreatic CSC phenotypes were expressed heterogeneously ranging from 0 to 99.35% (median 3.46) for ESA^+^CD24^+^CD44^+^ and 0 to 1.94% (median 0.13) for CXCR4^+^CD133^+^. Cell lines were classified according to ESA^+^CD24^+^CD44^+^ expression as: Low-Stemness (LS; <5%, n = 9, median 0.31%); Medium-Stemness (MS; 6–20%, n = 4, median 12.4%); and High-Stemness (HS; >20%, n = 4, median 95.8%) cell lines. Higher degree of stemness was associated with in vivo tumorigenicity but not with in vitro growth kinetics, clonogenicity, and chemo-resistance. A wide characterization (chemokine receptors, factors involved in pancreatic organogenesis, markers of epithelial–mesenchymal transition, and secretome) revealed that the degree of stemness was associated with *KRT19* and *NKX2.2* mRNA expression, with CD49a and CA19.9/Tie2 protein expression, and with the secretion of VEGF, IL-7, IL-12p70, IL-6, CCL3, IL-10, and CXCL9. The expression of stem cell markers was also evaluated on primary tumor cells from 55 PDAC patients who underwent pancreatectomy with radical intent, revealing that CXCR4^+^/CD133^+^ and CD24^+^ cells, but not ESA^+^CD24^+^CD44^+^, are independent predictors of mortality.

## 1. Introduction

PDAC is a lethal disease with a 5-year survival rate of approximately 10% and with a dramatic clinical course [1]. The adverse clinical outcome of PDAC is primarily due to the difficulty of an early diagnosis and the aggressiveness of cancer that is able to invade, disseminate, and resist to conventional therapies [2]. The latter was suggested to be driven by the existence of a population of highly plastic “stem”-like cells within the tumor, known as CSCs [3]. CSCs describe a subpopulation of cancer cells that behave like stem cells in their ability to self-renew and differentiate into different tumor components through stemness pathways [4,5]. The hypothesis that tumors may possess a stem cell–like subpopulation involved in driving tumor propagation and pathogenesis was suggested for many different solid tumors [5]. Since pancreatic CSCs were discovered in 2007 [3], different studies have demonstrated and confirmed their metabolic, invasive, and chemoresistance properties [6]. Pancreatic CSCs are involved in the mechanisms of tumor recurrence [7], drug resistance [8], tumor invasion, and metastatic dissemination [6]. Different strategies have been suggested to identify and isolate CSCs from pancreatic cancer including the expression of surface markers, the exclusion of the dye Hoechst 33342 (side population), and the ability to expand into spheres in suspension [9]. Even though a global signature of molecular markers does not exist yet [10], CSCs are known to express specific cell membrane markers. The tumor cells which simultaneously expressed CD44, CD24, and ESA/EpCAM (epithelial specific antigen) were firstly defined as pancreatic CSCs by Li et al. [3]. In fact, the CD44^+^CD24^+^EpCAM^+^ phenotype was reported to exhibit a 100-fold increase in the tumor-initiating capacity versus the other cancer cells, both in vitro and in vivo. Other suggested pancreatic CSC markers included CD133 [11], c-Met [12], ALDH1 [13], Lgr5 [14], DCLK1 [15], CXCR4 [16], and ABCG2 [17]. Among these, the double positivity for CD133 and CXCR4 was suggested to be able to identify a population of pancreatic CSCs that sustains pancreatic tumor growth and is essential for metastasis [18]. Even if the CSC concept is well accepted and believed to be the best model to understand PDAC heterogeneity and plasticity, some questions remain open [6], such as if CSC is a hardwire defined entity or a plastic state [19], and it is essential to develop methodologies for distinguishing CSCs from other tumor cells in order to identify new therapies that can specifically target and eliminate this cellular subpopulation, preventing tumor recurrence and metastatic disease. Established cell lines represent a commonly used source of material for PDAC studies, as the tumor’s conspicuous desmoplastic stroma makes the isolation of primary cells difficult. While data regarding genetic alterations, growth features, differentiation status, and biological behaviour in PDAC cell lines have been reported [20,21,22], a comprehensive analysis of stemness in commonly used PDAC cell lines has not been performed. In this report, we present the results of the analysis of the expression of stem cell markers in 17 pancreatic cancer cell lines and their association with growth kinetics, clonogenicity, chemo-resistance, tumorigenicity, expression of genes involved in pancreatic organogenesis and tissue commitment, expression of chemokine receptors, markers of epithelial–mesenchymal transition, and secretome. Moreover, the expression of stem cell markers in primary tumor from PDAC patients and their correlation with clinical outcome is provided.

## 2. Results

### 2.1. Evaluation of Putative Stem Phenotypes in PDAC Cell Lines

We evaluated the two putative stem phenotypes CD44^+^/CD24^+^/ESA^+^ and CD133^+^/CXCR4^+^ by flow cytometry analysis in 17 different PDAC cell lines (Figure 1 and Appendix A). The percentage of CD44^+^/CD24^+^/ESA^+^ cells was fairly heterogeneous between different cell lines ranging from 0 to 99.35% with a median of 3.46% (Figure 1A). The percentage of CD133^+^/CXCR4^+^ cell was generally low ranging from 0 to 1.94 % with a median of 0.13%. PDAC cell lines were classified according to CD44^+^/CD24^+^/ESA^+^ percentage as Low-Stemness (LS; <5%, n = 9, median 0.31%), Medium-Stemness (MS; 6–20%, n = 4, median 12.4%), and High-Stemness (HS; >20%, n = 4, median 95.8%) cell lines (Figure 1A,B). As expected, although with low percentages, the CD133^+^/CXCR4^+^ cells segregate according to the defined degree of stemness (Figure 1B). The analysis of all combinations of staining (Figure 1C and Appendix A) indicated that CD24^+^, CD133^+^, and CD24^+^/CD133^+^ cells (but not ESA^+^, CD44^+^, and CXCR4^+^) are correlated with the different degrees of stemness. Even if not statistically different, higher degrees of stemness are more represented in cell lines derived from distant metastases than in cell lines derived from loco regional tumoral tissue (Figure 1D), while patient’s age and sex were not different.

### 2.2. Growth Kinetics, Clonogenicity, Chemosensitivity, and Tumorigenicity according to the Degree of Stemness

All the 17 cell lines grew as a monolayer of substrate-adherent cells. Population doubling times ranged from 21 to 62 h and no correlation with the different degrees of stemness was evident (Figure 2A). We also performed a cell-cycle kinetics analysis and, consistently with the doubling time data, stemness was not associated with a different distribution of cell lines in the different phases of the cell cycle, nor with a different G2/G1 ratio, nor with a different percentage of apoptosis (SubG1) or hyperploidy (Figure 2B). In vitro clonogenicity of all the 17 cell lines was evaluated by means of IC50 (average number of cells to be seeded in a well of a 96-well plate to obtain the growth of a cell clone in half of the seeded wells). Clonogenicity ranged from IC50 0.02 (MiaPaCa-2) to IC50 724.5 (Panc-2) and no correlation with the different degrees of stemness was evident (Figure 2C). To assess the chemoresistance, the different cell lines were cultured for 6 days with gemcitabine (GEM) (1–1000 µM) and the dose-response curves were used to calculate the IC50, the drug concentration required to achieve 50% cell death. No significant association was evident between the stemness degree and the GEM resistance (Figure 2D). The ability of each PDAC cell line to generate tumors in vivo was investigated by performing limiting dilution subcutaneous implantation in CD1 nude mice. As previously reported [23], tumors formed with all cell lines (17 out of 17, 100%) after subcutaneous xenotransplantation of 1 × 10^6^ cells and in a proportionately smaller number of cell lines as the number of cells decreases: 14 out of 17 (82.4%) after the injection of 1 × 10^5^ cells, 2 out of 17 lines (11.8%) after the injection of 1 × 10^4^ cells, and 1 out of 17 (5.9%) after the injection of 1 × 10^3^ or 1 × 10^2^ cells. A Cox regression analysis identified both the number of cells injected and their stemness as independent factors for tumor engraftment with hazard ratios of 5.2 for any log in cell number increase (95%CI: 3.3–8.3, *p* < 0.001) and 1.6 (95%CI: 1–2.42, *p* = 0.048) for any degree increase in stemness. Considering the injection of 1 × 10^5^ cells (Figure 3A), the mean time of tumor appearance was 78 (95%CI: 54–106), 49.5 (95%CI: 31–67), and 12 (95%CI: 10–14) days for LS, MS, and HS, respectively (*p* < 0.001). The increase in the number of cells injected (1 × 10^6^ cells) generally reduced the mean time of tumor appearance but the difference between the different degrees of stemness remained significant [LS: 15 days (95%CI: 8–22); MS 5 days (95%CI: 4–6); HS 4 days (95%CI: 4–5); *p* = 0.002]. 

### 2.3. Gene Expression of Factors Involved in Pancreatic Organogenesis and Differentiated Tissues Commitment according to the Degrees of Stemness

Pancreatic CSCs could derive from adult cells de-reprogrammed to a ground state. For this reason, we evaluated the expression of genes involved in pancreatic organogenesis and tissue commitment. The mRNA expression profile of markers of pancreatic terminal cell fate (*KRT19, CHGA, CHGB, SYP, INS, GCG, GAD2, HNF1b, SOX17, NES, SNAI1*, and *VIM*) or transcription factors involved in embryonic pancreatic development (*PDX1, NKX6.1, ISL1, NEUROD1, NGN3, NKX2.2, ONECUT1, PAX4, PAX6,* and *PTF1a*) was evaluated by qRT-PCR (Figure 3B). The degree of stemness resulted associated with an increased expression of *KRT19* and decreased expression of *NKX2.2*. A trend was also evident for an increased expression of *PDX1, ISL1, HNF1b, PAX6,* and *NES*.

### 2.4. Expression of Factors Relevant to the Biological Behavior of Cancer Cells according to the Degrees of Stemness

The expression of chemokine receptors in tumors is correlated with different types of relapses or more or less favorable clinical outcomes [24,25,26]. We analyzed by flow cytometry the expression of CCR and CXCR chemokine receptors in all the 17 PDAC cell lines (Appendix A). The expression of the receptors was heterogeneous among the different lines. Five chemokine receptors were consistently (median >10%) expressed in the majority of the cell lines: CCR4, CCR5, CXCR1, CXCR3, and CXCR6. None of the investigated receptors resulted statistically associated with the degree of stemness, even if a trend was evident for CXCR3 and CX3CR1 (Figure 4A). The expression of other factors potentially relevant for the biological behavior of the cancer cell was investigated (Appendix A) including cell adhesion molecules (CD15, CD49a, CD49e, CD318, E-cadherin, CA19.9), receptors for growth factors (insulin, CD220; IGF-1, CD221; angiopoietin, Tie2; stem cell factor, CD117), modulators of microenvironment (CD142, CD200), markers of mesenchyme (CD73, CD166, CD105, CD31), and mesenchymal stemness (Stro-1, CD34). As for the chemokine receptors, the expression was heterogeneous among the different cell lines. A consistent expression of CD220, CD49a, CD49e, CD318, CD19-9, CD142, CD73, CD166, E-cadherin, and CD105 was present in the majority of the cell lines. CD49a and CA19.9/Tie2 were significantly associated with the degree of stemness, and a trend was evident for CA19.9/CD220, CA19.9/CD221, and E-Cadherin.

### 2.5. Evaluation of Secretome according to the Degrees of Stemness

Secretome was examined after 24 h of culture of all 17 PDAC cell lines (Table 1). Only proteins detectable in at least half of the cell lines were included in the analysis. The high-abundance secreted proteins (>1 ng/mL/10^6^ cell/24 h) included VEGF, CXCL1/GROalpha, CXCL8/IL-8, PDGF-BB, and MIF. VEGF was significantly associated with the degree of stemness (Figure 4B). A trend was also evident for CXCL1/GROalpha and CXCL8/IL-8, driven more by MS than by HS. Among the intermediate-abundance secreted proteins (1000–50 pg/mL) IL-7, IL-12p70, and IL-6 were significantly associated with the degree of stemness, and a trend was evident for IFNg and IL-1RA. Among the low-abundance secreted proteins (<50 pg/mL) CCL3/MIP-1α, IL-10, and CXCL9/MIG were significantly associated with the degree of stemness, and a trend was evident for IL1a, TNFa, and IL-9.

### 2.6. Evaluation of Putative Stem Phenotypes in PDAC Patients

PDAC tissue [0.41 (0.23–0.66) grams] was obtained from *55* patients who underwent pancreatectomy with radical intent. The tumor tissues were enzymatically and mechanically processed in order to isolate single cells for FACS analysis. The procedure was successful in 47 out of 55 patients [3.1 × 10^6^ (1–26) cells/gram of tissue]. The CXCR4^+^CD133^+^ phenotype was evaluated in 46 patients and the percentage of expression ranged from 0.02 to 3.97% with a median of 0.77% (0.3–1.33). The ESA^+^CD24^+^CD44^+^ phenotype was evaluated in 41 patients and the percentage of expression ranged from 0 to 9.81% with a median of 1.87% (0.77–2.63). Since the two phenotypes were not related (see Figure 5A), each was analyzed separately for its clinical impact. Tumors were divided in two groups according to median of ESA^+^CD24^+^CD44^+^ or CXCR4^+^CD133^+^ cell percentage [high stemness tumor (HST) > median, low stemness tumor (LST) ≤ median]. Patients and tumor characteristics according to tumor stemness are reported in Table 2. We estimated the overall survival according to tumor CXCR4/CD133 stemness: medians were 355 (212–497) days and 1004 (210–1797) days for patient with HST (n = 22) or LST (n = 24), respectively (Log Rank, *p* = 0.008; Breslow *p* = 0.019; see Figure 5B). The disease-free survival was not statistically different between the two groups. In the univariate Cox analysis, after adjusting for age and sex, we found a significant association between death and tumor grading [2.94 (1.3–6.7); *p* = 0.01], adjuvant CT/RT [0.2 (0.06–0.67); *p* = 0.009], HST [2.81 (1.23–6.24); *p* = 0.011], % of CXCR4^+^CD133^+^ cells [1.73 (1.13–2.6); *p* = 0.007], % of CXCR4^+^CD133^−^ cells [1.09 (1.008–1.18); *p* = 0.03], and % of CXCR4^+^ cells [1.09 (1.008–1.18); *p* = 0.03]. The multivariate analysis (Figure 6) confirmed tumor grade, adjuvant CT/RT, and HST (or CXCR4^+^CD133^+^ cells but not CXCR4^+^CD133^−^ and CXCR4^+^ cells) as independent predictors of PDAC mortality. We also estimated the overall survival according to tumor ESA/CD24/CD44 stemness: medians were 412 (104–720) days and 778 (335–1221) days for patient with HST (n = 20) or LST (n = 21), respectively (Log Rank, *p* = 0.58; Breslow *p* = 0.066; see Figure 5B). The disease-free survival was not statistically different between the two groups. In the univariate Cox analysis after adjusting for age and sex, we found a significant association between death and pN1 [2.7 (1.14–6.2); *p* = 0.022], percentage of lymph nodes positive for tumor localization [1.15 (1.003–1.33); *p* = 0.045], adjuvant CT/RT [0.15 (0.037–0.67); *p* = 0.56], % of ESA^+^CD24^+^CD44^−^ cells [1.043 (1.001–1.087); *p* = 0.046], % of ESA^−^CD24^+^CD44^−^ cells [1.036 (1.006–1.068); *p* = 0.02], and % of CD24^+^ cells [1.034 (1.011–1.057); *p* = 0.004]. The multivariate analysis (Figure 6) confirmed pN1, adjuvant CT/RT, and % of CD24^+^ cells (or ESA^−^CD24^+^CD44^−^ but not ESA^+^CD24^+^CD44^−^ cells) as independent predictors of PDAC mortality.

## 3. Discussion

In this study, we analysed the characteristics of 17 established pancreatic ductal carcinoma cell lines and we tried to correlate them with their grade of stemness. In particular, we analysed the hallmarks of the aggressiveness of this malignancy such as growth kinetic, clonogenicity, chemoresistance, tumorigenicity, expression of genes involved in pancreatic organogenesis and tissue commitment, expression of chemokine receptors, markers of epithelial–mesenchymal transition, and secretome. Even if a general correlation between the two major putative stemness phenotypes and all functional parameters for stemness could not be detected, the study has increased our understanding of the fundamental nature of this tumor. First of all, we systematically evaluated the two putative pancreatic cancer stem cells phenotypes ESA^+^CD24^+^CD44^+^ and CXCR4^+^CD133^+^ in all cell lines. Two relevant results emerged: (I) the expression among cell lines is heterogeneous; (II) the expression within lines is heterogeneous. Both of these two aspects were not taken for granted. By definition, established cell lines show self-renewal ability and tumor-initiating capacity, two constitutive characteristics of CSCs. One would therefore expect a homogeneously high expression of CSC markers in all cell lines. The fact that this is not the case suggests that the different stem cells markers identify specific and phenotypically defined sub population within multiple cell populations with the ability to form tumors and self-renew. This supports the hypothesis that different populations of CSC exist and the next challenge will be to understand how specific CSC populations are related to one another and whether each possesses specific functional properties [27]. Moreover, established cancer cell lines should consist of homogeneous/monoclonal cell populations. The fact that not all cells within a line express the same levels of stemness markers supports the hypothesis that CSC is not a fixed entity but rather represent a plastic state, as recently suggested [19]. A second relevant finding obtained by this systematic study on the stem cells phenotypes in PDAC cell lines is that few, but potentially relevant, features of tumor cells may be related to the degree of stemness. The different stem cells phenotypes consistently exhibited in vitro comparable growth kinetic, clonogenicity, and chemosensitivity to GEM, which constitute three important parameters for CSC [28,29]. This is in part unexpected as previous data suggested that pancreatic CSCs may be resistant to chemotherapy. Shah and colleagues [30] from M.D. Anderson showed that pancreatic cancer cell lines that selectively grew in culture media containing therapeutic doses of gemcitabine increased in expression of the stem cell markers CD24, CD44, and ESA. In a separate study, Hermann and colleagues [18] found that CD133 populations in the pancreatic cancer cell line were enriched after exposure to gemcitabine. Moreover, Lee et al. has observed that treatment with ionizing radiation and the chemotherapeutic agent gemcitabine results in enrichment of the CD44^+^CD24^+^ESA^+^ population in human primary pancreatic cancer xenografts [31]. Our results on the 17 cell lines show that, even though several cell lines displayed a resistance towards GEM, the more resistant lines did not segregate on the basis of the different grade of stemness. It is important to underline that our study did not evaluate the chemoresistance between different populations in the same line, but between different lines according to their stemness and this could partly explain the difference in results. However, when we assessed the ability of the lines to generate in vivo tumors, we observed that the stemness degree was independently associated with the tumor engraftment and growth, supporting the hypothesis that the relationship with the microenvironment may play a key role in bringing out the properties of CSCs. We studied systematically the expression of genes involved in pancreatic organogenesis and, interestingly, the degree of stemness was significantly related to an increased expression of *KRT19* and a decreased expression of *NKX2.2*. Notably, *KRT19* overexpression demonstrated to be associated with carcinogenesis, progression, and poor prognosis in PDAC patients becoming a valuable biomarker for PDAC prognosis [23,32]. In agreement with our findings on PDAC, *KRT19* was previously described as a marker of hepatocellular carcinoma CSCs with stem cell characteristics and tumor-initiating ability [33]. On the other hand, *NKX2.2* overexpression has reported to suppress the self-renewal capability of glioblastoma-initiating cells and its downregulation in vivo accelerates the tumor formation [34]. The relevance of some chemokine receptors, such as CX3CR1, CXCR4, and CXCR3, has been described in pancreatic cancer. The CX3CR1 receptor, expressed by PDAC cells, is involved in the mechanisms of neurotropism, one of the major causes of local relapse [24]. The CXCR4/CXCL12 axis seems to play an important role in the desmoplastic reaction characterizing PDAC [16], while CXCR3/CXCL10 resulted expressed in pancreatic tumor tissue [35], and their presence has been correlated with poor prognosis [36]. We thus investigated the expression of chemokine receptors in the 17 PDAC cell lines. The expression of the receptors was heterogeneous among the different lines and CCR4, CCR5, CXCR1, CXCR3, and CXCR6 resulted consistently expressed in the majority of the lines. None of the investigated receptors segregated with the degree of stemness, even if a trend was reported for CXCR3 and CX3CR1. Among the other proteins evaluated, CD49a and CA19.9/Tie2 resulted associated with the stemness of the cell. In agreement with our findings, CD49a previously demonstrated to be a biomarker that promotes therapy resistance and metastatic potential in pancreatic cancer [37]. Moreover, Tie2 was reported to be expressed in brain tumor and prostate cancer stem cells and was associated with the malignant progression of these tumors [38,39,40]. Pancreatic cancer cell lines have a characteristic ability to secrete soluble factors [21]. VEGF, CXCL1/GROalpha, CXCL8/IL8, PDGF-BB, and MIF resulted the most secreted proteins. Interestingly, all the above-mentioned molecules are pro-angiogenic factors. VEGF (the vascular endothelium growth factor) promotes the growth of new vessels as well as PDGF-BB (growth factor derived from BB isoform platelets). GRO and IL-8 belong to the family of chemokines with the ELR+ motif (the amino acids Glu-Leu-Arg) and which promote angiogenesis by binding to the CXCR2 receptor on the endothelium [41]. Moreover, MIF (migration inhibition factor) is a powerful chemoattractant of endothelial cells [42]. VEGF expression was significantly associated with the degree of stemness, and a trend was evident for CXCL1/GROalpha and CXCL8/IL-8, even if driven more by the medium-stemness lines than by the high-stemness ones. All these three factors were described to be related with pancreatic cancer aggressiveness [43,44,45]. Among the less secreted proteins, IL-7, IL-12p70, IL-6, CCL3, IL-10, and CXCL9 resulted significantly associated with the degree of stemness. Al these factors were described to be relevant for pancreatic cancer aggressiveness by a direct or indirect protumorigenic action which includes upregulation of neovasculogenesis, chemo/radio resistance, and local and/or systemic immune regulation [46,47,48,49]. Finally, the expression of stem cell markers was also evaluated on primary tumor cells from PDAC patients. As for the cell lines, the expression within primary tumor cells was heterogeneous. The clinical outcome revealed that that the CXCR4^+^/CD133^+^ and CD24^+^ but not ESA^+^CD24^+^CD44^+^ cell frequencies are independent predictors of mortality. This is in agreement with previous results suggesting a relationship between a lower survival rate and the expression of CD133 [50,51], CD24 [52], and CXCR4 [53,54]. All these three molecules, beyond being able to be involved in tumor differentiation and acquisition of stemness [18,54,55,56,57,58], were described to be involved in the regulation of the epithelial to mesenchymal transition [58], the invasiveness and metastatization of the tumor [58,59], and the resistance to hypoxia [55], apoptosis [60], and chemotherapy [61,62]. In conclusion, our work shows that some characteristics of pancreatic cancer cell lines are related to their degree of stemness. As the various cell lines revealed a great deal of diversity, we suggest viewing cautiously generalized interpretations of the results of in vitro studies with few cell lines. Due to the difficulty to obtain primary culture from the resected pancreatic cancer, the reported data might be used to create a biological catalogue of pancreatic cancer cell lines listing their specific stemness.

## 4. Materials and Methods

### 4.1. Cell Lines

The PDAC cell lines investigated are reported in Appendix A [20,22]. The human PDAC cell lines were derived from human primary tumor (n = 10: Panc-1, PT45, MIAPaCa-2, SKPC-1, BI, PC, Panc-2, PaCa-3, PaCa-44, and BxPc-3), lymph node metastases (n = 2: Hst-766 and T3M4), liver metastasis (n = 2: Capan-1 and CFPAC-1), or ascites (n = 3: A8184, HPAF-II and AsPC-1). All the cell lines were cultured in RPMI 1640 (Lonza, Swiss, Basel) supplemented with 10% fetal bovine serum (Lonza, Swiss, Basel), 1% penicillin/streptomycin, and 2 mM L-glutamine. Cells were maintained under standard culture conditions (5% CO_2_, 95% air in humidified chamber at 37 °C). All the cell lines were trypsinized when they reached 80–90% confluence and plated in a new 75 cm^2^ polystyrene flask (Corning, Glendale, AZ, USA) at the density of 1 × 10^6^ cells/flask. The immortalized epithelial cell line derived from non-tumor human pancreatic ducts HPDE6-E6E7 (HPD6E) [63] was kindly provided by Dr. Ming-Sound Tsao (University of Toronto, Toronto, ON, Canada). Human islet (HI) were purified from the pancreas of multi-organ donors as previously described [64].

### 4.2. Cytofluorimetric Analysis of Surface Markers

The expression of surface markers was evaluated by flow cytometry. When cells reached 80–90% confluence, they were trypsinized, suspended (2–4 × 10^5^ cells/sample) in 10%RPMI, and incubated for 1 h before the staining to allow the cells to re-express the proteins possibly damaged by trypsin treatment. For chemokine receptors detection, the cell culture medium was changed 2 h before detaching the cells to promote the membrane expression of receptors. Cells were washed and incubated for 30 min at 4 °C with the primary antibody of interest appropriately diluted. The antibodies used for the staining are reported in Appendix A. The antibodies anti CA19-9 and anti Stro-1 were not directly conjugated to a fluorochrome. The Zenon^®^ technology (Life Technologies, Carlsbad, CA, USA) was used to non-covalently bind the Alexa-488 fluorochrome to anti CA19-9 following the instructions in the protocol attached by the manufacturer. Cy^TM^5-conjugated AffiniPure F(ab’)_2_ Fragment Goat anti-mouse IgM, **μ** chain specific, was used as a secondary antibody for the anti Stro-1. Labeled cells were analyzed in a BD FACSCanto II (diva software) or BD facscalibur (cellquest software) cytometer (Becton Dickinson, NJ, USA). The results were expressed as percentage of positive cells. Unlabeled cells were used as a control. Representative plots of the gating strategy of the stem phenotypes CD44^+^/CD24^+^/ESA^+^ and CD133^+^/CXCR4^+^ by flow cytometry are presented in Appendix A.

### 4.3. Growth Kinetics and Cell-Cycle Analysis

To determine the proliferative capacity of the cell lines, 2 × 10^4^ cells per well were plated in duplicate in 24-well plates in standard medium at day 1. The number of cells was counted every about 40 h after plating until day 10 using a Burker counting chamber. The growth curve was drawn and the cell doubling time during logarithmic growth was calculated according to the standard formula [65]: Doubling time = Length Log (2)/[ Log (Final concentration) − Log (Initial concentration)]. For the cell-cycle analysis, cells were marked with propidium iodide (solution of 1 μL of PI at a concentration of 1 μL/mL, 5 μL RNAse at a concentration of 0.1 mg/mL, 194 μL PBS) and the percentage of cells in the subG1, G1, S, and G2/M phases quantified by FACS. The analysis was carried out with the program FCS express v3 (De Novo Software, Los Angeles, CA, USA).

### 4.4. Clonogenicity Assay In Vitro

Cells were seeded at different concentrations (0.1, 0.5, 1, 3, 5, 10, 100 cells/well) in a 96-well plate (Corning, Glendale, AZ, USA). Twenty-four replicates were performed for each concentration point. After 10 days, the percentage of the positive wells, presenting clones, was calculated on the total of the 24 wells under the optical microscope (10X and 5X magnification). A clone is defined as a colony made up of at least 4 cells. The clonogenicity was expressed as IC50: average number of cells to be seeded in a well of a 96 plate to obtain the growth of a cell clone in half of the seeded wells. The IC50 was calculated using the nonlinear quadratic minima curve using the CalcuSyn program (Biosoft, Oxford, UK).

### 4.5. Chemoresistance Assay

PDAC cell lines were seeded at a concentration of 1 × 10^5^ cells/mL per well, in a 12-well polystyrene cell culture plate (Corning, Glendale, AZ, USA). The cells were treated at different doses of GEM (1, 10, 100, 1000 µm). After 6 days, cells were labelled with PI and the DNA content measured by flow cytometry. The dose-response curves were then plotted and the drug concentrations inducing an apoptosis of 50% (IC50) were calculated. The IC50 was calculated using the nonlinear minimum quadratic curve using the CalcuSyn program (Biosoft, Oxford, UK).

### 4.6. Tumorigenic Capacity in Mice

Increasing doses (1 × 10^2^, 1 × 10^3^, 1 × 10^4^, 1 × 10^5^, 1 × 10^6^ cells/mouse) of cells for each PDAC cell line were transplanted heterotopically in athymic CD1 nude mice (male, 8 weeks old, Charles River). Cells were washed 3 times and injected in D-PBS1X (100 μL total volume) subcutaneously. Three mice per dose were inoculated for each cell line. Health conditions and the appearance of xenografts were monitored three times a week, for a follow-up period extended up to 196 days after injection. The date of first tumor formation was recorded when a mass of approximately 3 × 3 mm^3^ was palpable. Mice were housed in specific pathogen-free conditions in accordance with the guidelines of the San Raffaele Scientific Institute Animal Care and Use Committee.

### 4.7. Gene Expression Analysis by Real Time PCR

Gene expression analysis was performed as previously described [23]. Briefly, total (1–5 μg) RNA was isolated with TRIZOL (Invitrogen, Waltham, MA, USA) and reverse transcribed using the SuperScript III RT kit (Invitrogen, Waltham, MA, USA), according to the manufacturer’s instructions. Quantitative real time PCR (qRT-PCR) was performed using predesigned gene-specific TaqMan Gene Expression Assays (Applied Biosystems, Waltham, MA, USA, listed in Appendix A) in a 7900 Real-Time PCR System (Applied Biosystems, USA). The relative expression levels of each gene were calculated with the 2 ∆∆Ct method using the GAPDH gene as endogenous control and expressed as arbitrary units (AU) corresponding to fold changes relative to a human islet (HI, dashed line) preparation used as reference tissue (AU = 1). The value AU = 1E-09 was assigned to genes not detected by qRT-PCR.

### 4.8. Secretome by Luminex Xmap Technology

PDAC cell lines were cultured at a concentration of 1 × 10^6^ cells per ml in RPMI 10% in a 25 cm^2^ (Corning, Glendale, AZ, USA) polystyrene flask. After 24 h of culture, the secreted factors were measured by a multiplex bead-based assay based on Luminex technology (Bio-Plex Pro™ Human Cytokine; Bio-Rad, Milan, Italy). The panel included: IL-1 beta (IL-1b), Interleukin-1 receptor antagonist (IL-1RA), IL-2, IL-4, IL-5, IL-6, IL-7, IL-8/CXCL8, IL-9, IL-10, IL-12 (p70), IL-13, IL-15, IL-17, CCL11/eotaxin, basic Fibroblast Growth Factor (bFGF), Granulocyte-Colony Stimulating Factor (G-CSF), Granulocyte-Macrophage Colony-Stimulating Factor (GM-CSF), IFN-gamma, CXCL10/IP-10, CCL2/MCP-1, CCL3/MIP1-alpha, CCL4/MIP1-beta, CCL5/RANTES, Tumor Necrosis Factor alpha (TNF-a), IL-1 alpha (IL-1a), Interleukin 2 Receptor alpha (I*L2RA*), IL-3, IL-12 (p40), IL-16, IL-18, CCL27/CTAK, CXCL1/Gro-alpha, Hepatocyte Growth Factor (HGF), Interferon alpha 2 (IFN-a2), Leukaemia Inhibitory Factor (LIF), CCL7/MCP-3, Macrophage Colony-Stimulating Factor (M-CSF), Macrophage Migration Inhibitory Factor (MIF), CXCL9/MIG, Nerve Growth Factor beta (NGF), Stem Cell Factor (SCF), Stem Cell Growth Factor beta (SCGF-b), CXCL12/SDF-1, Tumor Necrosis Factor beta (TNF-b), TNF-related apoptosis-inducing ligand (TRAIL), Platelet-derived Growth Factor (PDGF-BB), and Vascular Endothelial Growth Factor (VEGF).

### 4.9. Cohort of PDAC Patients and Tissue Processing

From January 2008 to March 2010, 55 adult patients were randomly selected among cases of PDAC admitted to the Pancreatic Surgery Unit at San Raffaele Hospital. The characteristics of patients are described in Appendix A. The local IRB approved the study, and all patients provided a written informed consent. Tumor tissue was obtained from pancreatic neoplasms after surgical resection. A small piece of the tumor was digested to single cells (Cancer Cell Isolation Kit, Panomics Inc., Reedwood City, CA, USA) and cultured in RPMI10% before staining for flow cytometry analysis.

### 4.10. Statistical Analysis

Variables are shown as mean ± standard deviations (SD) or as median and interquartile ranges, according to their distribution. Variables with a normal distribution were compared with one-way unpaired (two groups) or one-way ANOVA test (three or more groups). Variables with a non-normal distribution were compared with Wilcoxon signed-rank test (two groups) or Kruskal–Wallis test (three or more groups). Categorical variables were compared with the chi-square test or Fisher’s exact test, as appropriate. Disease-free and overall survival were estimated according to Kaplan–Meier. Association between variables were evaluated by Cox regression analysis. All statistical analyses were performed using the SPSS statistical software, version 13.0 (SPSS Inc, Chicago, IL, USA).

## Figures and Tables

**Figure 1 ijms-23-10663-f001:**
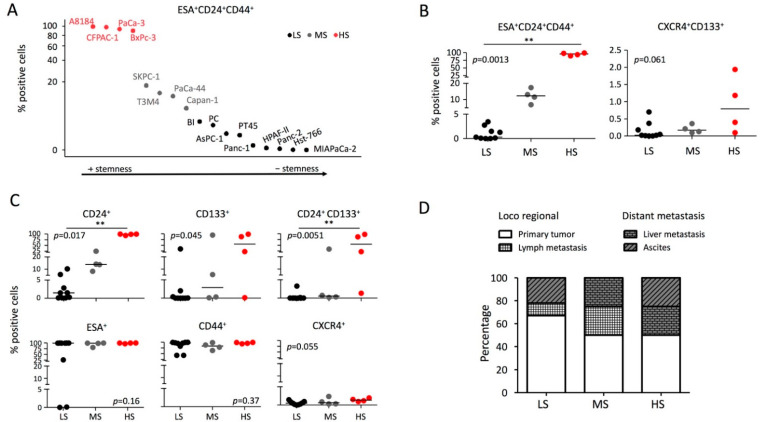
**PDAC cell lines phenotype by FACS analysis.** (**A**) Expression of the stem phenotypes ESA^+^/CD24^+^/CD44^+^ in pancreatic cancer cell lines. According to ESA^+^/CD24^+^/CD44^+^ percentage, the lines were classified as Low-Stemness (LS; <5%, n = 9), Medium-Stemness (MS; 6–20%, n = 4), and High-Stemness (HS; >20%, n = 4) cell lines. (**B**) Percentage of ESA^+^/CD24^+^/CD44^+^ and CD133^+^/CXCR4^+^ cells according to the different stemness degrees. (**C**) Expression of CD24, CD133, CD24/CD133, ESA, CD44, and CXCR4 according to the different stemness degrees. (**D**) Tissue origin of PDAC cell lines according to the different stemness degrees. The Kruskal–Wallis test with post hoc Dunn’s multiple comparison test was applied. ** *p* < 0.05. The medians are represented as horizontal lines.

**Figure 2 ijms-23-10663-f002:**
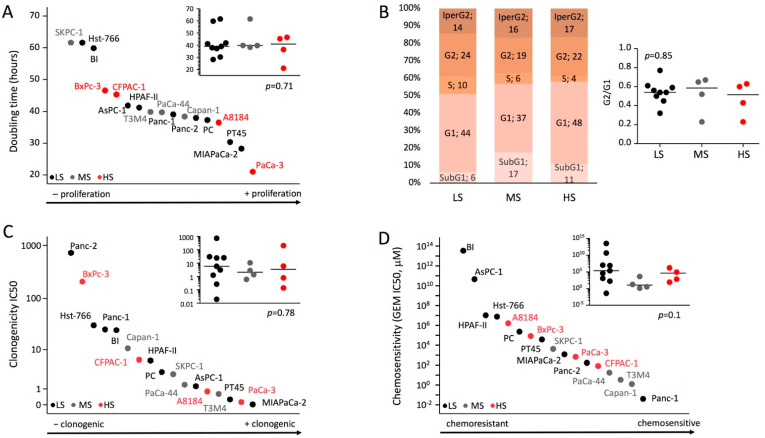
**Growth, clonogenicity, and chemosensitivity of the PDAC cell lines according to the different stemness degrees.** (**A**) Scatter dot plot representing the doubling time (hours) of the different cell lines. (**B**) Cell cycle distribution of the cell lines and ratio G2/G1 according to the different stemness degrees. (**C**) Clonogenicity IC50 in the different cell lines. (**D**) Gemcitabine IC50 (μM) to show the chemoresistance after 6 days treatment with gemcitabine (1–1000 µM) of the different cell lines. The Kruskal–Wallis test with post hoc Dunn’s multiple comparison test was applied. The medians are represented as horizontal lines.

**Figure 3 ijms-23-10663-f003:**
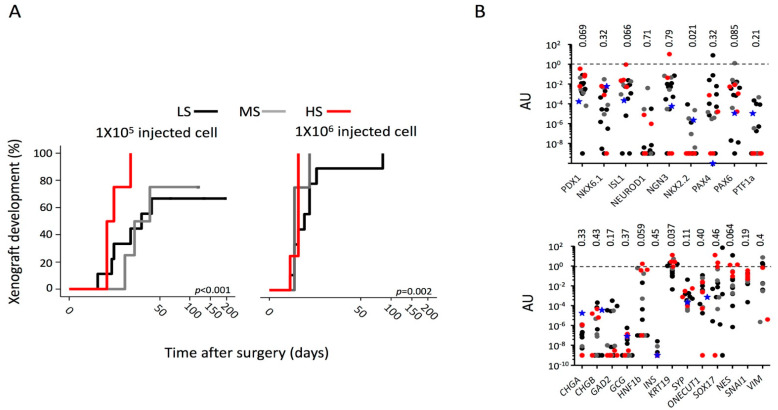
**Tumorigenicity of PDAC cell lines and gene expression profile according to the different stemness degrees.** (**A**) Kaplan–Meier curve estimating the cumulative probability to develop xenograft of the 17 PDAC cell lines injected in CD1 nude mice according to the different stemness degrees. Shown is the *p*-value estimated by the log-rank test. (**B**) Gene expression analysis of transcription factors involved in embryonic pancreatic development and markers of terminal cell fate by qRT-PCR. Each dot represents a line. Black dot: Low-Stemness lines; Grey dot: Medium-Stemness lines; red dot: High-Stemness lines; blue star: normal human pancreatic duct cell line HPDE6-E6E7. The Kruskal–Wallis test with post hoc Dunn’s multiple comparison test was applied.

**Figure 4 ijms-23-10663-f004:**
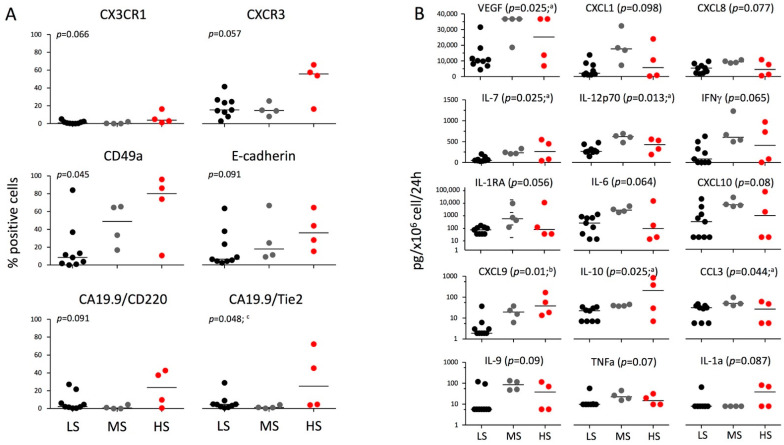
**Expression of factors relevant for the biological behavior and secretome according to the different stemness degrees.** (**A**) Percentage of expression of CX3CR1, CXCR3, CD49a, E-cadherin, CA19.9/CD220, and CA19.9/Tie2 in the 17 PDAC lines, divided according to the different stemness degrees: LS (CD44^+^/CD24^+^/ESA^+^ < 5%, n = 9), MS (CD44^+^/CD24^+^/ESA^+^ 6–20%, n = 4), and HS (CD44^+^/CD24^+^/ESA^+^ > 20%, n = 4). (**B**) PDAC cell lines were cultured at a concentration of 1 × 10^6^ cells/mL for 24 h and secreted factors were measured by a multiplex bead assay based on Luminex technology. Represented are the factors which resulted differentially expressed according to the different stemness degrees at *p* < 0.1 (see Table 1). The Kruskal–Wallis test with post hoc Dunn’s multiple comparison test was applied: a: *p* < 0.05 LS vs. MS; b: *p* < 0.05 LS vs. HS; c: *p* < 0.05 MS vs. HS. Black dot: Low-Stemness lines; Grey dot: Medium-Stemness lines; red dot: High-Stemness lines. The medians are represented as horizontal lines.

**Figure 5 ijms-23-10663-f005:**
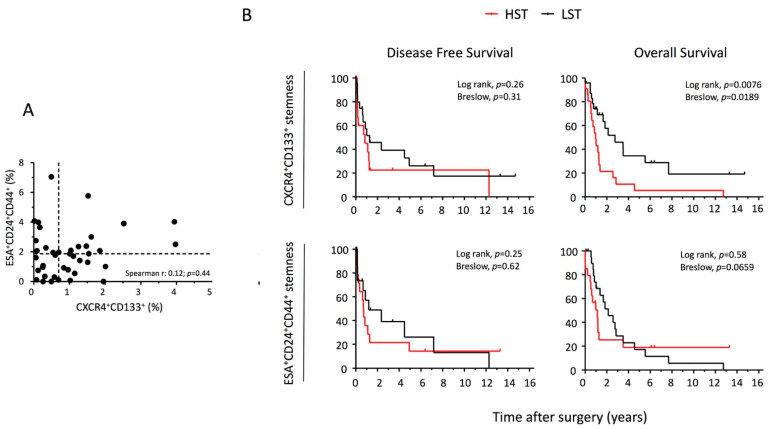
**PDAC tissue stemness phenotype and overall/disease-free survival.** Tumor tissues from 47 patients were enzymatically and mechanically processed in order to isolate single cells for FACS analysis. (**A**) Correlation between ESA^+^CD24^+^CD44^+^ and CXCR4^+^CD133^+^ cell percentage. Spearman’s rho statistic was applied. Dotted lines represent the median values. (**B**) Kaplan–Meier curves representing the overall survival and disease-free survival. Tumors were divided in two groups according to median of ESA^+^CD24^+^CD44^+^ or CXCR4^+^CD133^+^ cell percentage [high stemness tumor (HST) > median, low stemness tumor (LST) ≤ median]. Statistical analysis was performed by log-rank test and Breslow test.

**Figure 6 ijms-23-10663-f006:**
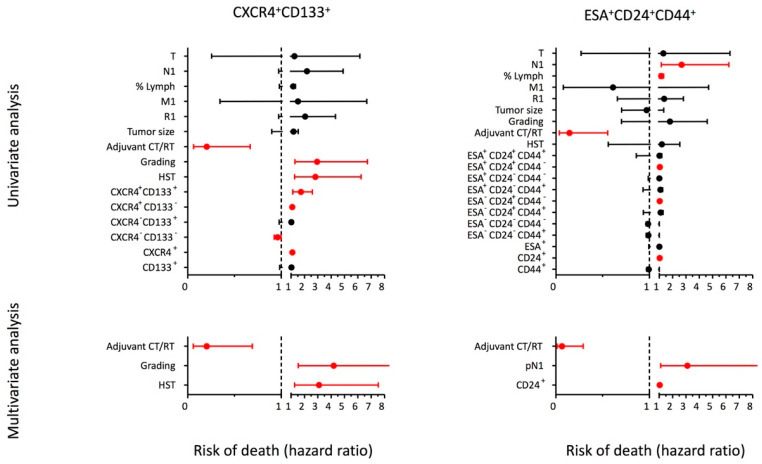
**Univariate and multivariate hazard ratios for PDAC mortality.** The associations between baseline variables, tumor stemness according to CXCR4^+^CD133^+^ (left panels, n = 46) or ESA^+^CD24^+^CD44^+^ cell percentage (right panels, n = 41), and PDAC mortality were assessed by Cox regression analysis. All analyzed variables are presented. Dots represent the hazard ratio after lines of the 95% confidence intervals. Red dot/lines *p* < 0.05. CT/RT: chemo/radiotherapy; HST: high stemness tumor (see the text).

**Table 1 ijms-23-10663-t001:** Evaluation of secretome according to the different stemness degrees.

	All 17 lines	N	LS	N	MS	N	HS	N	*p*
	pg/mL/24 h		pg/mL/24 h		pg/mL/24 h		pg/mL/24 h		
**High-abundance secreted proteins >1000 pg/mL**									
VEGF	13,749 (8978–36,740)	0	10,112 (7488–14,906)	0	36,741 (23,193–36,741)	0	25,245 (8570–36,741)	0	0.025
CXCL-1	7264 (8978–36,740)	0	2072 (1217–13,779)	0	17,593 (9661–28,839)	0	5736 (4435–20,622)	0	0.098
CXCL8	6917 (2272–9328)	0	5420 (2272–7628)	0	9365 (8738–10,404)	0	4539 (766–9999)	0	0.077
PDGF	1700 (466–3457)	0	1536 (463–2324)	0	4832 (1885–6697)	0	1225 (63–2474)	0	0.12
MIF	1637 (773–3332)	0	1173 (454–3460)	0	3151 (1573–3360)	0	1554 (976–2382)	0	0.59
**Intermediate-abundance secreted proteins 1000—50 pg/mL**									
CXCL12	964 (739–1266)	0	965 (601–1759)	0	993 (776–1056)	0	821 (202–998)	0	0.52
SCGF-b	821 (540–1745)	0	660 (499–1857)	0	866 (609–1054)	0	1261 (551–18,671)	0	0.81
CXCL10	619 (9.5–3537)	6	158 (9–1550)	4	3538 (2829–11,420)	0	492 (9–29,443)	2	0.08
IL-6	372 (16.2–1209)	3	149 (14–441)	2	1605 (1142–3008)	0	55 (9–6508)	1	0.044
IFN-g	329 (6.4–647)	5	86.9 (6–414)	4	603 (511–1089)	0	410 (27–913)	1	0.065
CCL5	326 (143–1624)	0	270 (147–1310)	0	826 (370–14,137)	0	143 (60–2292)	0	0.18
IL-12p70	325 (254–540)	0	267 (251–379)	0	621 (51–677)	0	426 (224–548)	0	0.013
LIF	108 (75–434)	0	163 (75–373)	0	289 (73–929)	0	107 (42–426)	0	0.88
M-CSF	90 (25–665)	3	88 (18.4–692)	2	370 (144–1779)	0	691 (49–896)	1	0.25
SCF	85 (39–199)	0	91 (51–199)	0	59.5 (24–84)	0	244 (76–1007)	0	0.14
IL-7	78 (48–229)	0	52 (42–105)	0	230 (211–310)	0	263 (53–522)	0	0.028
IL-1RA	59 (20–166)	6	41 (20–64)	4	318 (111–4053)	0	44 (20–4648)	2	0.056
IL-13	59 (48–70)	0	55 (43–64)	0	71(68–78)	0	56 (43–69)	0	0.12
**Low-abundance secreted proteins <50 pg/mL**									
G-CSF	45 (6–124)	3	20.5 (5.6–84.4)	2	82.9 (41–5061)	0	362 (7.7–1741)	1	0.29
IL-4	13.2 (3.6–19.5)	4	11 (4.6–15.4)	2	18 (8.5–35)	0	12 (1.3–29)	2	0.48
CCL3	38 (5.7–48)	5	31.4 (5.7–38.5)	3	49.7 (42.4–85.8)	0	26.5 (5.7–57.3)	2	0.044
IL-10	30 (7.1–39.7)	5	22.7 (7.1–32)	4	40 (37.7–44)	0	208 (13–736)	1	0.025
CXCL9	6.1 (1.9–30)	5	1.9 (1.9–4.6)	5	19.5 (8.5–34)	0	38 (15–141)	0	0.01
IL-18	6.1 (3.4–8.4)	5	6.1 (3.4–8.4)	3	6.2 (3.9–7.04)	1	16.9 (3.6–273)	1	0.78
IL1a	8.7 (6.4–236)	6	6.4 (6.4–97.5)	5	163 (22–854)	0	9.7 (6.9–530)	1	0.087
CCL2	26.5 (7.2–973)	8	7.2 (7.2–689)	6	325 (44–6346)	0	28 (7.2–1083)	2	0.18
TNF-a	9.7 (9.7–23)	9	9.7 (9.7–10)	7	22.6 (16.3–40.1)	0	15 (9.7–29)	2	0.073
HGF	8.3 (8.3–13.8)	9	8.3 (8.3–13.8)	5	5.4 (8.3–10.5)	2	26.7 (8.3–90.4)	2	0.66
CCL27	6.9 (6.9–24)	9	6.9 (6.9–20)	5	6.9 (6.9–18.8)	3	57.1 (8.1–152)	1	0.28
IL-9	5.7 (5.7–104)	9	5.7 (5.7–49)	7	84 (48–128)	0	38 (5.7–105)	2	0.091
CCL4	5.6 (5.6–24.3)	9	5.6 (5.6–13.8)	6	22.6 (9.3–55.7)	1	14.6 (5.6–86.6)	2	0.25
CCL7	4.8 (4.8–94)	9	4.8 (4.8–86.6)	4	4.8 (4.8–4.8)	3	74.7 (4.8–239)	2	0.53
bNGF	4.1 (4.1–16)	9	4.1 (4.1–19.5)	7	9 (4.9–17.8)	0	6.1 (4.1–27.6)	2	0.28
IL-2Ra	2.3 (2.36–60)	9	48.4 (2.36–72.3)	3	2.3 (2.3–2.3)	4	29 (2.3–108)	2	0.14
**Under the limit of detection**									
CCL11	<8	10	-	8	-	0	-	2	nt
IL-15	<7.3	11	-	8	-	1	-	2	nt
GM-CSF	<3	11	-	8	-	1	-	2	nt
IL-12p40	<8.2	13	-	7	-	4	-	2	nt
IL1b	<10.7	14	-	8	-	3	-	3	nt
IL-5	<10	14	-	8	-	4	-	2	nt
FGFb	<4.8	14	-	9	-	1	-	4	nt
IL-16	<3.8	14	-	8	-	4	-	2	nt
IL-3	<7.9	14	-	8	-	4	-	2	nt
IL-2	<5.7	15	-	9	-	3	-	3	nt
IFNa2	<0.86	15	-	9	-	4	-	2	nt
TNF-b	<6.8	16	-	9	-	4	-	3	nt
TRAIL	<6.3	16	-	9	-	4	-	3	nt
IL-17	<7.4	17	-	9	-	4	-	4	nt

LS = low stemness; MS = medium stemness; HS = high stemness; *p* = *p* value; N = number of cell lines in which the protein was under the limit of detection.

**Table 2 ijms-23-10663-t002:** Patients and tumor characteristics according to tumor stemness.

	CXCR4/CD133			ESA/CD24/CD44		
	HST (n = 22)	LST (n = 24)	*p*	HST (n = 20)	LST (n = 21)	*p*
**Patients characteristics**						
Age (years; mean ± sd)	57.7 ± 10.5	59.5 ± 9.4	0.87	67.5 ± 9.1	65 ± 10.5	0.45
Sex (M/F)	12/10	14/10	0.8	13/7	8/13	0.12
Neo-adjuvant CT [n, (%)]	3 (13.6)	0 (0)	0.1	2 (10)	1 (4.8)	0.61
Adjuvant CT/RT [n, (%)]	13/19 (68.4)	16/20 (80)	0.48	11/17 (64.7)	17/17 (100)	0.018
Overall survival (median)	355d	1004d	0.008 * 0.019 ^§^	412d	778d	0.58 * 0.07 ^§^
Disease free survival (median)	307d	465d	0.26	255d	435d	0.62
Local Relapse [n, (%)]	6 (27.3)	5 (20.8)	0.73	1 (5)	7 (33.3)	0.045
Distant relapse [n, (%)]	10 (45.5)	11 (45.8)	1	12 (60)	7 (33.3)	0.12
-Liver	4 (40)	9 (81.8)	0.08	6 (50)	6 (86)	0.17
-Lung	2 (20)	1 (9.1)	0.59	2 (17)	0 (0)	0.51
-Lymph nodes	4 (40)	4 (36.4)	1	3 (25)	4 (57)	0.33
-Peritoneal Carcinomatosis	2 (20)	2 (18.2)	1	1 (8.39	1 (14.3)	1
-Other sites	1 (10)	2 (18.2)	1	0 (0)	2 (28.6)	0.12
**Tumor characteristics**						
Tumor size (cm)	2.5 (2–3)	2.9 (2–3.9)	0.34	2.6 (2–3)	2.8 (2–3.5)	0.51
pT1 [n, (%)]	0 (0)	1 (4)	0.38	1 (5)	0 (0)	0.33
pT2 [n, (%)]	0 (0)	1 (4)		1 (5)	0 (0)	
pT3 [n, (%)]	22 (100)	22 (92)		18(90)	21 (100)	
pN1 [n, (%)]	18 (81.8)	15 (62.5)	0.197	14 (70)	15 (71.4)	1
Lymph nodes pos (%)	31 (11–50)	33 (19–41)	0.95	30 (12.5–71)	32 (17–45)	0.79
pM1 [n, (%)]	2 (9.1)	3 (12.5)	1	1 (5)	3 (14.3)	0.61
R1 [n, (%)]	12 (54.2)	13 (54.2)	1	10 (50)	13 (61.9)	0.54
Grading [n, (%)]:			0.39			1
G1	0 (0)	0 (0)		0 (0)	0 (0)	
G2	14 (64)	12 (50)		11 (55)	12 (57)	
G3	8 (36)	12 (50)		9 (45)	9 (43)	
CXCR4^+^CD133^+^	1.4 (1–1.9)	0.32 (0.1–0.5)	<0.001	1 (0.17–1.6)	0.7 (0.26–1.1)	0.4
CXCR4^+^CD133^−^	2.03 (1.2–8)	0.9 (0.26–3.1)	0.041	2.5 (1.3–6.5)	1.9 (0.21–4.2)	0.11
CXCR4^−^CD133^+^	4 (0.71–8.8)	2.36 (1–4.8)	0.38	2.8 (0.5–7.5)	1.8 (0.8–4.4)	0.75
CXCR4^−^CD133^−^	90 (82–95)	95 (93–98)	0.01	93 (82–95)	94 (86–98)	0.18
CXCR4^+^	3.5 (2.4–9.6)	1.5 (0.36–3.7)	0.0023	3.5 (2–9.5)	2.8 (0.77–4.8)	0.2
CD133^+^	4.8 (2.2–11.4)	2.7 (0.1–5.4)	0.06	3.8 (0.2–11.3)	2.97 (1.3–5.3)	0.52
ESA^+^CD24^+^CD44^−^	3.8 (0.62–7.9)	1.7 (0.34–3.7)	0.25	4.2 (1.4–8.3)	1.3 (0.21–3.7)	0.035
ESA^+^CD24^+^CD44^+^	1.9 (0.94–2.5)	1.7 (0.3–2.6)	0.52	2.6 (2.1–4)	0.8 (0.1–1.36)	<0.001
ESA^+^CD24^−^CD44^+^	0.9 (0.25–2.7)	2.04 (0.6–3.6)	0.15	1.96 (0.5–3.4)	0.8 (0.21–2.3)	0.12
ESA^+^CD24^−^CD44^−^	12.3 (7.4–36)	0.9 (0.25–2.7)	<0.001	14.9 (5.9–22)	8.7 (3–36)	0.82
ESA^−^CD24^+^CD44^−^	1.41 (0.3–6.2)	1.14 (0.2–2.1)	0.12	1.2 (0.3–3.1)	1.4 (0.05–3)	0.84
ESA^−^CD24^+^CD44^+^	1.36 (0.4–3)	0.14 (0.02–0.7)	0.005	1.1 (0.2–2)	0.1 (0.01–1)	0.027
ESA^−^CD24^−^CD44^−^	48 (38–63)	57 (36–70)	0.31	50 (28–65)	53 (36–68)	0.61
ESA^−^CD24^−^CD44^+^	13 (6.9–18.4)	12.3 (4.7–18)	0.55	14.6 (8.7–18)	6.6 (3–18.5)	0.09
ESA^+^	18 (5–36)	22 (11–44)	0.35	25 (16–38)	14 (5–44)	0.26
CD24^+^	11.5 (5.9–31)	6.3 (3.4–13.8)	0.08	10 (6.5–17.2)	5.2 (2–14.2)	0.030
CD44^+^	19.2 (9–24.6)	13.6 (5–22.5)	0.27	21 (19–26)	11.2 (5.6–21)	0.013

HST = high stemness tumor; LST = low stemness tumor; *p* = *p* value; * Log rank analysis, ^§^ Breslow analysis.

## Data Availability

Not applicable.

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
