# Peer review of "A Comprehensive Characterization of Stemness in Cell Lines and Primary Cells of Pancreatic Ductal Adenocarcinoma"

_ijms, 2022, doi:10.3390/ijms231810663_

Round 1
Reviewer 1 Report
In this study, Ferrara et al have analyzed the characteristics of 17 established pancreatic ductal carcinoma cell lines and have tried to correlate them with their grade of stemness. They have analyzed the hallmarks of the aggressiveness of this malignancy like growth kinetic, clonogenicity, chemoresistance, tumorigenicity, expression of genes involved in pancreatic organogenesis and tissue commitment, expression of chemokine receptors, markers of epithelial–mesenchymal transition, and secretome. The analysis of expression of stem cell markers in 17 pancreatic cancer cell lines and their association with growth kinetics, clonogenicity, chemo-resistance, tumorigenicity, expression of genes involved in pancreatic organogenesis and tissue commitment, expression of chemokine receptors, markers of epithelial–mesenchymal transition, and secretome. They also have provided the expression of stem cell markers in primary tumor from PDAC patients and their correlation with clinical outcome.
Overall, the study is well conducted, and all the experiments were performed in systemic manner. The results well support the proposed claims, and the results are well described. There are some comments which will further improve the manuscript. I would recommend a minor revision before the editors’ s decision.
Comments
1. Figure 3A the legends are missing in both X and Y axis.
2. The graph in Figure 3 B should be described in detail. Which cell line was used for this analysis? Were all the 17 cell lines tested?
3. In figure 3 B The degree of stemness resulted associated with an increased expression of KRT19 and decreased expression of NKX2. However, CGC shows similar trend as NKX2 in the expression. Please discuss this.
4. In the line 131 -136 the authors described that the injection of 1*105 cells (Figure 3A), the mean time of tumor appearance was 12 (95%CI: 10-14), 49.5 (95%CI: 31-67) and 78 (95%CI: 54-132 106) days for LS, MS, and HS, respectively (p<0.001). For (1*106 cells) the mean time of tumor appearance were [LS: 15 days 135 (95%CI:8-22); MS 5 days (95%CI:4-6); HS 4 days (95%CI:4-5); p=0.002]. Please check the mean time in 1*105 cells as it is reflected here that the mean time is more in HS then LS.
5. In addition to the gold standard in vivo dilutional tumor propagation assays used to identify CSCs; CSCs have also been identified based on in vitro sphere forming assays. Experiment to demonstrate self-renewal capacity in vitro such a tumor sphere assay will be important.
6. In figure 4 it is not clear which groups were statistically compared. y
7. Previous data suggests that pancreatic CSCs may also be resistant to chemotherapy and radiation. Shah and colleagues from M.D. Anderson showed that pancreatic cancer cell lines that selectively grew in culture media containing therapeutic doses of gemcitabine demonstrated morphologic and biochemical properties of epithelial-to-mesenchymal transition. In a separate study, Hermann and colleagues found that CD133 populations in the pancreatic cancer cell line were enriched after exposure to gemcitabine. Also, Cheong J. Lee et al has also observed that treatment with ionizing radiation and the chemotherapeutic agent gemcitabine results in enrichment of the CD44+CD24+ESA+ population in human primary pancreatic cancer xenografts. The authors should discuss this in their paper
8. The survival analysis were compared between LS and HS. Did the author analyzed the survival for MS
Author Response
Please see the attachmen

Reviewer 2 Report
1. In figure 3A, what dose X axis mean? what dose Y axis mean?
2. Authors should provide IRB license Number..
Author Response
|
Figure 3A the legends are missing in both X and Y axis. |
As suggested, legends were included |
|
Authors should provide IRB license Number |
Number and Title of IRB is now reported |
Reviewer 3 Report
The manuscript by Benedetta Ferrara and coworkers investigated on the expression of cancer stem cell (CSC) markers in 17 PDAC cell lines and in primary tumor cells from 55 PDAC patients. The two putative stem phenotypes CD44+/CD24+/ESA+ and CD133+/CXCR4+ were evaluated. Higher degree of stemness was not associated with in vitro growth kinetics, clonogenicity and chemoresistance. However, the stemness degree was independently associated with the tumor engraftment and growth. The topic is interesting but there are a few points that need to be addressed.
Main points:
- The authors do not provide representative plots of the gating strategy of the stem phenotypes CD44+/CD24+/ESA+ and CD133+/CXCR4+ by flow cytometry for PDAC cell lines and primary tumor cells. This data should be provided at least in the supplementary materials. Moreover, the authors state “Unlabeled cells were used as a control”. Was the mean fluorescence intensity (MFI) ratio values calculated by dividing the MFI of positive events by the MFI of negative events? Please, explain better.
- For the clonogenicity assay in vitro, it is not described by which method the cell clones were evaluated. Furthermore, it is difficult to think about seeding 0.1, 0.5 or 1 cells/well in a96-well plate. Please, provide more information.
- To assess the chemoresistance, the different PDAC cell lines were cultured for 6 days with gemcitabine (GEM) (1-1000 μM) and the dose-response curves were used to calculate the IC50. Figure 2D shows very high IC50 values for some cell lines, a supplementary table of IC50 values would be useful. How are the IC50 values compared to those found in the literature and calculated by MTT assay after treatment with gemcitabine?
Minor points:
- There are some typos throughout the text. For example: some sentences are in italics (lines 17 and 27); some references could be merge (for example line 343).
